# Risk Assessment of CHD Using Retinal Images with Machine Learning Approaches for People with Cardiometabolic Disorders

**DOI:** 10.3390/jcm11102687

**Published:** 2022-05-10

**Authors:** Yimin Qu, Jack Jock-Wai Lee, Yuanyuan Zhuo, Shukai Liu, Rebecca L. Thomas, David R. Owens, Benny Chung-Ying Zee

**Affiliations:** 1Division of Biostatistics, The Jockey Club School of Public Health and Primary Care, Faculty of Medicine, The Chinese University of Hong Kong, Hong Kong SAR, China; 1155101052@link.cuhk.edu.hk (Y.Q.); jack@cuhk.edu.hk (J.J.-W.L.); 2Department of Acupuncture and Moxibustion, Shenzhen Traditional Chinese Medicine Hospital, Shenzhen 518005, China; qinyuan64@163.com; 3Department of Cardiovascular Disease, Shenzhen Traditional Chinese Medicine Hospital, Shenzhen 518005, China; lsk23456@163.com; 4Diabetes Research Group, Swansea University, Swansea SA2 8PP, UK; r.l.thomas@swansea.ac.uk (R.L.T.); owensdr@cardiff.ac.uk (D.R.O.); 5Clinical Trials and Biostatistics Lab, CUHK Shenzhen Research Institute, Shenzhen 518057, China

**Keywords:** coronary heart disease, retinal images, machine learning, cardiometabolic disorders

## Abstract

Background: Coronary heart disease (CHD) is the leading cause of death worldwide, constituting a growing health and social burden. People with cardiometabolic disorders are more likely to develop CHD. Retinal image analysis is a novel and noninvasive method to assess microvascular function. We aim to investigate whether retinal images can be used for CHD risk estimation for people with cardiometabolic disorders. Methods: We have conducted a case–control study at Shenzhen Traditional Chinese Medicine Hospital, where 188 CHD patients and 128 controls with cardiometabolic disorders were recruited. Retinal images were captured within two weeks of admission. The retinal characteristics were estimated by the automatic retinal imaging analysis (ARIA) algorithm. Risk estimation models were established for CHD patients using machine learning approaches. We divided CHD patients into a diabetes group and a non-diabetes group for sensitivity analysis. A ten-fold cross-validation method was used to validate the results. Results: The sensitivity and specificity were 81.3% and 88.3%, respectively, with an accuracy of 85.4% for CHD risk estimation. The risk estimation model for CHD with diabetes performed better than the model for CHD without diabetes. Conclusions: The ARIA algorithm can be used as a risk assessment tool for CHD for people with cardiometabolic disorders.

## 1. Introduction

Globally, coronary heart disease (CHD) is the leading cause of death and affected 110 million people and accounted for 8.9 million deaths in 2015 [1]. Coronary heart disease caused approximately one third of all deaths in persons older than 35 years of age [2]. The prevalence and mortality of CHD vary according to risk factors among countries and regions and change markedly over time. Although CHD prevalence has been decreasing in many developed countries, the overall disease burden is still increasing, especially in low- and middle-income countries, due to rapid socioeconomic development and aging [3,4,5,6,7,8,9]. Approximately three fourths of the global mortality and 82% of the global disability-adjusted life years (DALYs) of CHD occurred in low- and middle-income countries [10]. China has undergone a rapid health transition in recent decades, with rapid economic growth and social change [11]. A crude prevalence of 4.42% was reported in rural China in 2019 [12]. The global burden of disease 2015 study showed that the age-standardized CHD mortality rate increased by 13.3% from 1990 to 2015 in China [13]. The mortality rate was 113.46 per 100,000 for Chinese urban residents and 118.74 per 100,000 for rural residents according to the “China Health and Family Planning Statistical Yearbook 2017” [14]. A study predicted a 69% increase in CHD events and a 64% increase in CHD-related deaths during 2020–2029 compared to 2000–2009 [15]. Among the CHD patients who died within one month after the onset of symptoms, approximately two thirds died before they reached a hospital after suffering a CHD event [16].

Cardiometabolic disorders, including hypertension, type 2 diabetes, overweightness or obesity, and dyslipidemia, are documented preclinical factors associated with cardiovascular diseases [17,18,19]. People suffering from cardiometabolic disorders are at higher risk of developing CHD in the future than the general population [20,21,22,23,24,25,26]. Therefore, risk assessment of CHD among these patients has more clinical significance.

The tools used for the diagnostic assessment of CHD, such as coronary angiography and computed tomography (CT), are reliable and well established. Sophisticated imaging modalities can determine the presence and severity of the diseases with a high degree of sensitivity and specificity. However, these diagnostic tools are technically challenging and expensive to operate, and thus unsuitable as a screening tool in the general population, especially in rural areas. Thus, feasible and cost-efficient tests must be developed to enable the diagnostic identification of individuals with a high risk of CHD.

Many risk equations have been developed in recent years to predict the risk of CHD in the general population, such as those described in the Framingham Study, the Atherosclerosis Risk in Communities (ARIC) Study, the Prospective Cardiovascular Münster (PROCAM) Study, the CUORE cohort, and the Taiwan cohort study [27,28,29,30,31,32,33,34]. Risk equations for diabetes patients were also established, such as the Cardiac Risk Calculator from the UKPDS study [35]. These risk equations included similar clinical characteristics which substantially vary in disease risk. However, a proportion of disease morbidity and mortality cannot be explained by these risk factors [36,37,38,39]. Additionally, these models’ practice value in primary care settings is limited since they require information from questionnaires, invasive blood tests, and physical examinations [28,29,30,31,40]. Moreover, it is more challenging to develop risk estimation models for people with cardiometabolic disorders since risk factors used in traditional risk equations such as blood pressure, blood glucose, and blood lipid levels were also abnormal for these people.

Thus, there is a pressing need for new, convenient, and inexpensive risk factors for CHD assessment and prevention for people with cardiometabolic disorders. We noted that retinal vessels are the only directly visible vessels in the body and microvascular retinopathy features are associated with coronary artery abnormalities [41,42,43]. Retinal vascular changes are a summary marker of a patient’s lifetime exposure to risk factors and can reflect cumulative vascular damage [44]. Furthermore, many studies found retinal changes associated with CHD and CHD mortality [45,46,47,48,49,50,51]. The analysis of retinal characteristics may provide a supplementary way for risk estimation of CHD.

## 2. Materials and Methods

### 2.1. Study Subjects

This cross-sectional study was conducted in the Cardiology Department of Shenzhen Traditional Chinese Medicine Hospital from December 2017 to September 2019. The Shenzhen Traditional Chinese Medicine Hospital Ethics Committee (Ref. No.: K2019-005-01) and the Joint Chinese University of Hong Kong—New Territories East Cluster Clinical Research Ethics Committee (Ref. No.: 2020.093) approved the study. Informed consent was obtained from each participant. The inclusion criteria used in the study included good health status and ability to sit on a chair for taking a retinal image, diagnosis with a cardiometabolic disorder, and having a clear disease diagnosis. Subjects with cataracts or other eye diseases that affected retinal imaging required close clinical monitoring or who were unable or unwilling to comply with disease examination were excluded.

Baseline demographic and medical information was collected from participants’ hospital medical records. The nurses measured height and weight and calculated body mass index (BMI; calculated as weight in kilograms divided by height in meters squared) for each participant. Diabetes mellitus (DM) was defined as a fasting blood glucose concentration > 7.0 mmol/L or an HbA1c value > 6.5%; patients with a history of diabetes were also labelled as having DM. Hypertension was defined as a systolic blood pressure ≥ 140 mmHg, diastolic blood pressure ≥ 90 mmHg, or a patient with a history of hypertension. Dyslipidemia was defined as total cholesterol (TC) ≥ 240 mg/L, triglyceride (TG) ≥ 200 mg/dL, high-density lipoprotein cholesterol (HDL-C) < 40 mg/dL, low-density lipoprotein cholesterol (LDL-C) ≥ 160 mg/dL, or a patient with a history of dyslipidemia. Obese status was defined as a BMI of 28 or higher, overweight status as a BMI of 24.0 to 28, and normal weight (including underweight) status as a BMI of less than 24.0. Cardiometabolic disorders, including hypertension, diabetes, dyslipidemia, and overweight or obese status, were measured based on the hospital clinical protocol. Patients with at least one of these conditions were recruited. CHD was diagnosed according to standard recommendations from international guidelines [52,53,54]. Patients were initially suspected of CHD based on their symptoms and electrocardiogram results. Coronary computed tomography or coronary angiography were performed for accurate diagnosis in accordance with clinical practice guidelines. The presence of 50% diameter stenosis of any coronary a with a diameter ≥ 2.0mm has been used as the threshold value for diagnosis.

Three hundred sixteen subjects were included in this study: 188 CHD patients and 128 control with cardiometabolic disorders.

### 2.2. Retinal Imaging Acquisition and Analysis

Retinal images of both eyes were taken within two weeks from the hospital admission using a non-mydriatic fundus camera (Canon CR2). Each participant obtained one retinal image centered at the fovea of each eye. The spatial resolution of each retinal image was 3648 by 2432 pixels, and the images were stored without compression before analysis.

A fully automatic retinal image analysis for CHD risk (ARIA-CHD) was developed using R (version 3.6.0, R Foundation for Statistical Computing, Vienna, Austria) and Matlab (Version 2020a, The Math Works, Inc., Natick, MA, USA) computer software to estimate retinal microvascular characteristics and incorporate machine learning techniques to estimate an overall risk of CHD. The detailed methods of the automatic retinal imaging analysis method have been reported [55]. The methods include fractal analysis, high order spectra analysis, and statistical texture analysis incorporating a machine learning approach. The characteristics include retinal vessel measurements, arteriole–venous nicking, arteriole occlusion, hemorrhage, exudates, tortuosity, bifurcation coefficients (BC), asymmetry of branches, bifurcation angles, and other machine-learning generated factors.

### 2.3. Statistical Analysis

For univariate analysis, independent t-test and chi-square test were used to evaluate if the retinal and clinical variables were significantly different between the CHD and control groups. Variables with a *p*-value of less than 0.05 were considered statistically significant.

For the classification analysis, we randomly selected 70% for the training of the classification model. The other 30% were used for an internal validation process. We used machine learning and deep learning techniques. We extracted the texture/fractal/spectrum-related features (such as high order spectra and fractal dimensions) associated with CHD by using the automatic retinal image analysis (ARIA) algorithm written in Matlab [44]. We then used the glmnet approach to select the most important subset of features based on the penalized maximum likelihood using R and Matlab [55]. Finally, we translated the features extracted from the machine learning approaches to commonly used retinal characteristics measured from the images using ImageJ to gain further insights. We have previously applied this method and validated results in different disease cohorts [56,57,58].

For the validation, we applied a 10-fold cross-validation method using the Support Vector Machine (SVM) algorithm to test the datasets that have not been used in the model’s training [59,60]. This is done by partitioning the dataset and using 90% of the data to train the algorithm and the remaining 10% of data for testing. Because cross-validation does not use all of the data to build a model, it is a commonly used method to prevent overfitting during training.

### 2.4. Sample Size Estimation

To obtain sensitivity and specificity values of 0.85 or higher with a lower bound of the 95% confidence intervals of at least 0.7, we require more than 50 subjects to estimate sensitivity and specificity for each subgroup [61,62].

## 3. Results

The univariate analysis showed that many characteristics were significantly different between the CHD patients and the control group (Table 1). The CHD group are older and has a higher rate of female participants than the cardiometabolic disorders group (*p* < 0.001). In addition, the percentage of patients with hypertension, diabetes, and dyslipidemia was higher in the CHD group than in the control group (*p* < 0.005).

The univariate analysis also revealed differences in many retinal characteristics between the CHD patients and people with cardiometabolic disorders (Table 2). We found significantly more tortuosity (*p* = 0.020 for average tortuosity, *p* = 0.013 for arteriole tortuosity, *p* < 0.001 for venule tortuosity, respectively), more exudates (*p* = 0.001 for left eyes), and more arteriole occlusion (*p* = 0.032 for left eyes) in the CHD group. While the central retinal artery equivalent (CRAE, *p* = 0.028 for right eyes), central retinal vein equivalent (CRVE, *p* = 0.002 for left eyes), mean bifurcation coefficient of venules (MBCV, *p* = 0.014 for left eyes), mean asymmetry index of arterioles (MAasymmetry, *p* < 0.001 for left eyes), the mean asymmetry index of venules (MVasymmetry, *p* = 0.008 for left eyes), and mean bifurcation angles of arterioles (MAangle, *p* = 0.007 for right eyes) were smaller in the CHD group.

After the univariate analysis, we randomly selected 70%—132 CHD and 90 controls—for training the classification model. The remaining 30%—56 CHD and 38 control—were used for internal validation. The results of this classification model are our primary analysis. Thirty-three controls (86.8%) and 52 CHD patients (92.9%) were correctly classified in the internal validation set. The ROC curve was displayed in Figure 1, with an AUC of 0.96. The overall accuracy was 90.4% (Figure 2).

We further divided CHD patients into diabetes and non-diabetes groups for subgroup analysis. The same training and testing procedures were conducted for the two subgroup analyses. The results showed that for classification between CHD without diabetes and control, the model had a sensitivity of 84.6% and a specificity of 89.5%, with an overall accuracy of 87.5%. For classification between CHD with diabetes and control, the model had a sensitivity of 90.0% and a specificity of 94.7%, with an overall accuracy of 92.6% (Figure 3).

To avoid overfitting and test the robustness of the models, we performed a 10-fold cross-validation analysis by using an SVM algorithm for testing datasets that were not used in the training of the model. This was performed by 10-fold partitioning the dataset and repeating the training and testing ten times. Each analysis used a subset of 90% of the data to train the algorithm and the remaining 10% for testing. This process was repeated ten times until all ten folds of the data were tested. We then calculated the overall sensitivity and specificity for the cross-validation analysis based on all ten independent analyses. The sensitivity and specificity results from the cross-validation analysis were 88.3% and 81.3%, respectively, for the overall CHD model; 90.6% and 84.1%, respectively, for CHD with diabetes; 89.1% and 89.0%, respectively, for CHD without diabetes.

## 4. Discussion

WHO has recommended cost-effective interventions that can be applied in low-resource settings to prevent CHD. The population-level prevention strategies include comprehensive tobacco control; reducing intake of foods high in fat, sugar, and salt; encouraging physical activity; reducing alcohol consumption; and promoting healthy diet habits. Individual-level prevention requires targeting people at high risk. These diseases can be prevented by addressing the above behavioral risk factors to target the high-risk population. Thus, early screening with appropriate tools is urgently needed.

The relationship between retinal images and cardiovascular diseases has been studied previously. Many retinal characteristics are associated with CHD risk factors, disease presence, and prognosis. For instance, retinal vessel atherosclerosis was correlated strongly with the risk factors and severity of CHD, and retinal arteriolar endothelial dysfunction predicts major adverse cardiovascular events in patients with CHD or cardiovascular risk factors [63,64,65]. Automatic retinal vessel analysis may provide added benefit to traditional risk factors in stratifying patients at risk for CHD.

Retinal vessel calibers are the most studied retinal characteristics. The association of CRAE, CRVE, and CHD have been reported previously, though the results were not consistent [66,67,68]. Our result showed that CHD patients had smaller CRAE and CRVE than the control group. Retinal vascular tortuosity quantifies the frequency with which a vessel crosses a low dimensional spline [69]. The tortuosity of vessels constitutes a physiological mechanism that increases metabolic tissue input [70]. Several studies have reported positive associations between retinal vessel tortuosity and cyanotic CHD, intracranial artery atherosclerosis, diabetes, and cerebrovascular events. However, one research study reported a negative association with ischemic heart disease death [68,71,72,73,74]. Our study result demonstrated a positive association between retinal vessel tortuosity and CHD. For retinal fractal dimension characteristics, we found a negative association of MAangle, MAasymmetry, and MVasymmetry with CHD. Similar associations have been found with cerebrovascular events and stroke previously [74]. We have also found larger occlusion and more exudates in the CHD group than the control group in this study, consistent with previous studies [75,76,77].

Our study shows that retinal images can be used as a risk assessment tool for CHD in people with cardiometabolic disorders. The sensitivity and specificity for the classification model were 92.9% and 86.8%, respectively. After dividing CHD patients into diabetes and non-diabetes groups, the model performed even better in the diabetes subgroup. The results of cross-validation were also robust.

Recently, one study using data from South Korea, Singapore, and the UK has shown that a deep learning retinal coronary artery calcium (CAC) score is comparable to CT-scan-measured CAC in predicting cardiovascular events [78]. Another study has established a risk assessment model using fundus photographs for a 10-year risk assessment of ischemic cardiovascular diseases in China, which proved the utility of retinal images as a risk assessment tool in the general population [79]. Compared with these studies, we chose people with cardiometabolic disorders as the control group since they are at higher risk of developing cardiovascular diseases. Though these people were more difficult to classify from CHD patients, the performance of our model was just as good, if not better.

However, there were several limitations to this research. First, although the retinal images were captured prospectively, this is still a retrospective study since CHD patients may have their disease developed when admitted to the hospital. Second, all the participants in this study were patients from the Shenzhen Traditional Chinese Medicine Hospital, and no community controls were recruited. Thus, we cannot eliminate potential case-selection bias in this research. Third, we did not have a separate and large enough data set for model testing for further external validation.

For the future direction of the study, we aim to explore the ability of automatically calculated retinal images to prospectively evaluate the risk of CHD by incorporating retinal images captured in prospective community-based cohorts. Population with cardiometabolic disorders can have regular retinal image acquisition and automated analysis, and subjects with a high burden of developing CHD can undergo further evaluation.

## Figures and Tables

**Figure 1 jcm-11-02687-f001:**
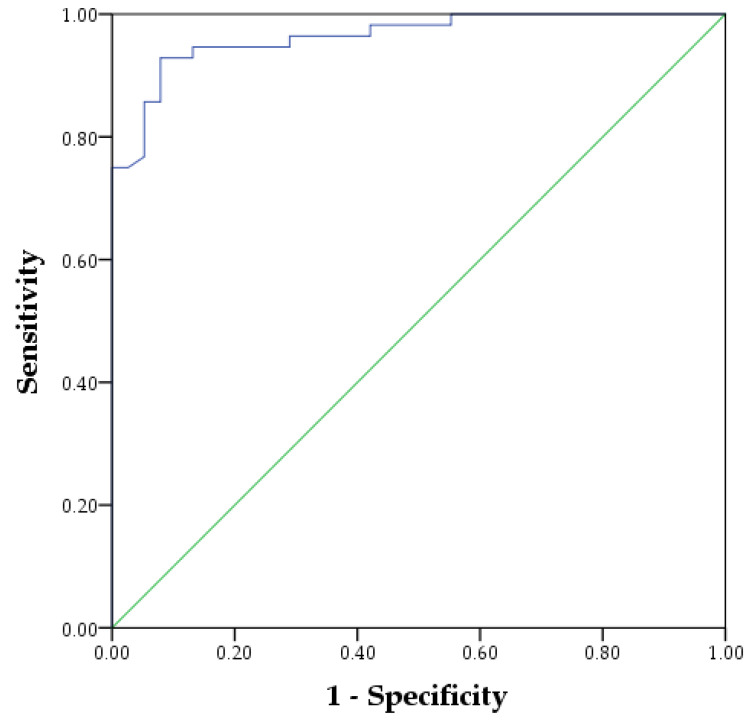
The ROC curve of the classification model for CHD.

**Figure 2 jcm-11-02687-f002:**
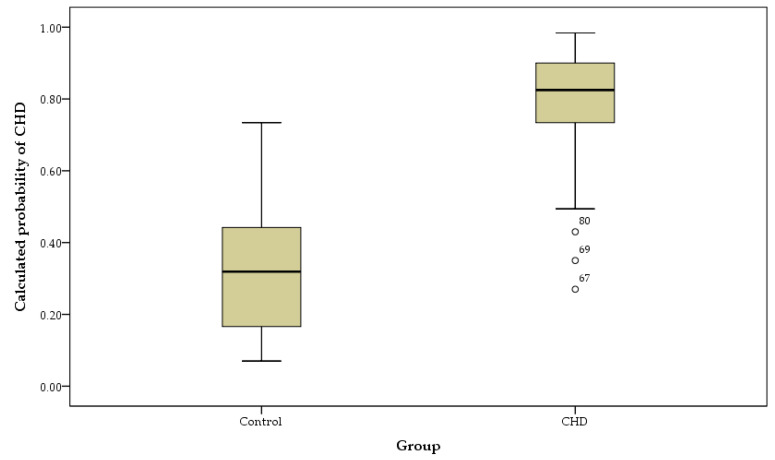
The classification model for CHD in box plot.

**Figure 3 jcm-11-02687-f003:**
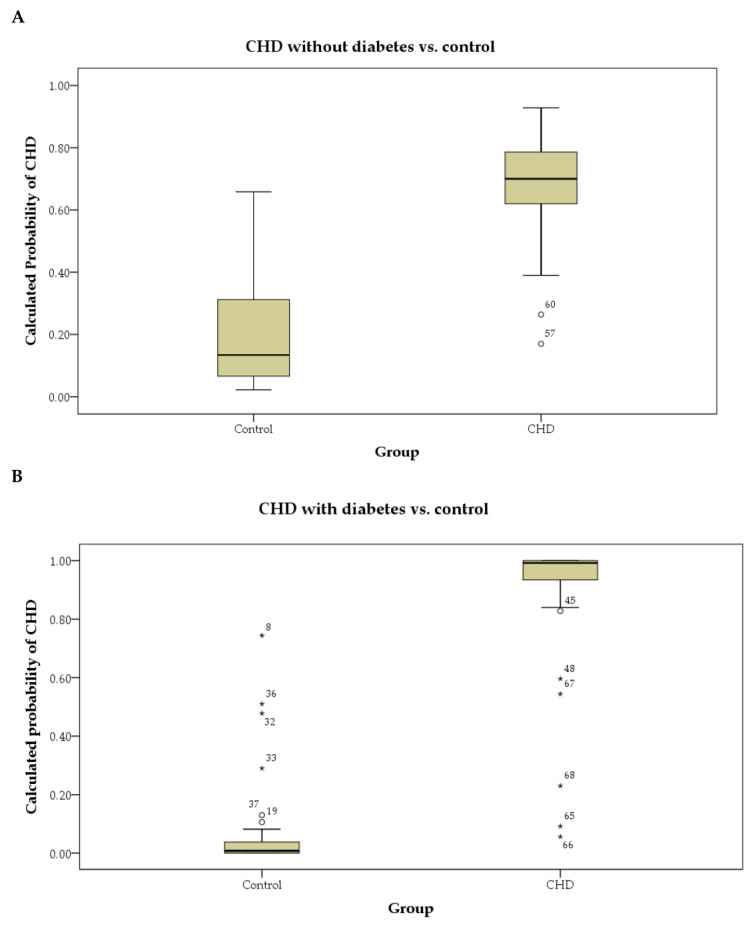
Subgroup analysis with classification models for CHD patients (**A**) without and (**B**) without diabetes in box plot.

**Table 1 jcm-11-02687-t001:** Patient characteristics between coronary heart disease (CHD) and cardiometabolic disorders.

Basic Characteristics	Control *n* = 128	CHD *n* = 188	*p*
Age (years)	52.13 ± 11.78	63.89 ± 11.40	<0.001
Sex *n*, (%)			<0.001
Male	42(32.81%)	103(55.79%)	
Female	86(67.19%)	85(45.21%)	
Smoking *n*, (%)			0.891
No	115(89.84%)	168(89.36%)	
Yes	13(10.16%)	20(10.64%)	
Drinking *n*, (%)			0.100
No	114(89.06%)	177(94.15%)	
Yes	14(10.94%)	11(5.85%)	
BMI group			0.200
<24	70(54.69%)	89(47.34%)	
≥24	58(45.31%)	99(52.66%)	
Diabetes *n*, (%)			<0.001
No	97(75.78%)	95(50.53%)	
Yes	31(24.22%)	93(49.47%)	
HbA1c (%)	6.25 ± 1.41	6.66 ± 1.26	0.019
Fasting glucose (mmol/L)	5.16 ± 2.16	5.63 ± 1.95	0.050
Hypertension *n*, (%)			<0.001
No	49(38.28%)	34(18.09%)	
Yes	79(61.72%)	154(81.91%)	
SBP (mmHg)	135.39 ± 22.05	133.87 ± 20.26	0.529
DBP (mmHg)	85.53 ± 14.39	80.64 ± 13.47	0.002
Dyslipidemia n, (%)			0.043
No	50(39.06%)	53(28.19%)	
Yes	78(60.94%)	135(71.81%)	
TG (mmol/L)	1.85 ± 1.34	1.90 ± 1.90	0.791
TC (mmol/L)	4.56 ± 0.98	4.32 ± 1.29	0.076
HDL-C (mmol/L)	1.20 ± 0.33	1.13 ± 0.31	0.073
LDL-C (mmol/L)	2.85 ± 0.90	2.67 ± 1.10	0.119

**Table 2 jcm-11-02687-t002:** Comparison of retinal characteristics between CHD and Control.

Retinal Characteristics	Control *n* = 128	CHD *n* = 188	*p*
lCRVE	18.34 ± 0.36	18.21 ± 0.38	0.002
lMBCV	1.21 ± 0.03	1.20 ± 0.03	0.014
lMAasymmetry	0.85 ± 0.01	0.85 ± 0.01	<0.001
lMVasymmetry	0.75 ± 0.01	0.74 ± 0.01	0.008
lAocclusion	0.13 ± 0.08	0.16 ± 0.09	0.032
lExudates	0.23 ± 0.07	0.26 ± 0.08	0.001
lTortuosity_av	0.20 ± 0.07	0.22 ± 0.08	0.020
lTortuosity_a	0.14 ± 0.06	0.16 ± 0.07	0.013
lTortuosity_v	0.15 ± 0.06	0.18 ± 0.08	<0.001
rCRAE	11.17 ± 0.26	11.10 ± 0.25	0.028
rMBCV	1.20 ± 0.02	1.20 ± 0.02	0.015
rMAangle	76.76 ± 1.44	76.32 ± 1.44	0.007

## Data Availability

Datasets are available upon request to the corresponding author.

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
