# Peer review of "Risk Assessment of CHD Using Retinal Images with Machine Learning Approaches for People with Cardiometabolic Disorders"

_jcm, 2022, doi:10.3390/jcm11102687_

Round 1

Reviewer 1 Report

The authors have addressed my previous comments. 

Author Response

Dear reviewer,

Thank you so much for your comments.

Best regards,

Yimin

Reviewer 2 Report

The authors failed to adequately and sufficiently address the concerns that were noted in the previous review. This is especially important in regards to the methodology of the study and the questions about coronary angiography and coronary CT, which have not been adequately answered. Additionally, the authors state that the reason for the admissions of research subjects were "suspected hypertension or heart disease". Those are not valid reasons for hospital admissions. Overall, the methodology of the study has significant flaws and unclear information.

Reviewer 3 Report

The manuscript has been significantly improved. All my questions have received answers. 

Author Response

Dear reviewer,

Thank you so much for your comments.

Best regards,

Yimin

This manuscript is a resubmission of an earlier submission. The following is a list of the peer review reports and author responses from that submission.

Round 1

Reviewer 1 Report

It was not clearly stated how the subjects were divided

into 2 groups? Was it all patients who have cardiometabolic

syndrome and then it was divided into those with CHD 

or not.

Reviewer 2 Report

Dear authors,

I am very glad to review your paper. 

The topic is very interesting andf the paper is well-written

The statistical analysis is appreciable and the results are very interesting despite the size of the sample is small.

I invite you to design a prospective study to endorse these results

Best Regards

Reviewer 3 Report

Minor and major comments:

- The authors did not define what is in their study CHD. This is a major limitation.

- Computed tomography (CT) is the gold standard to diagnose CHD in asymptomatic patients and is not technically challenging to use and is not expensive to operate. If CT is not available there are large number of clinical scores to detect CHD like the SCORE risk chart form ESC. This could be a major limitation.

- In this study were performed Coronary angiography or Coronary Computed Tomography for coronary heart disease diagnosis. Coronary angio in asymptomatic patient is not acceptable. Two completely different techniques.

Reviewer 4 Report

This is an interesting topic for a study that assessed the utility of retinal vasculature evaluation as a surrogate for coronary heart disease risk. The study is well-organized with a good flow of ideas and information. The authors also applied various statistical methods to analyze their data as mentioned in the study. There are major issues in the methodology of the study.

It is very unclear on how the study was actually performed for the following reasons.

First, it is mentioned in the study that the subjects were admitted to the hospital and that the retinal images were obtained within two weeks of the admission. Such an information cannot be laid out as such without further details on the reasons behind these admissions, i.e. what were the reasons these patients and the control subjects were admitted to the hospital?

Second, it is very unclear on what formed the basis for the selection of control group subjects. The authors mention that the control group constituted a group of subjects with cardiometabolic disorders, which included hypertension, diabetes, dyslipidemia, overweight or obesity. Nonetheless, we still do not know how these subjects were enrolled in the study. Were they admitted to the hospital between 12/2017 and 9/2019 (see the point above for the admission question)? If so, again why were those subjects admitted to the hospital to begin with? Even if they were seen in the outpatient clinic and not admitted, this needs to be clearly stated with a comprehensive elaboration on the process of selecting these subjects and enrolling them in the analysis. Additionally, how many of the cardiometabolic disorders listed were required in a single subject to be eligible for the study? This should be mentioned briefly for clarification.

Line 112-113: “Coronary angiography or Coronary Computed Tomography were performed for coronary heart disease diagnosis”

This is a very important point in the study and forms the basis of the categorization into CHD patients vs control subjects, yet there is nothing mentioned on the process of performing and evaluating the results of these studies. As this is a core point in the manuscript, the reader needs to know, and must be provided with the following information:

  • How many coronary angiographies were performed, and who were the patients who underwent this invasive study?
  • How many coronary CTs were performed, and who were the patients who had evaluation with this noninvasive imaging modality?
  • The reasons behind discrepancies in the method of evaluation, i.e. why did some patients had a coronary angiography while others had a coronary CT?
  • What were the indications for the invasive coronary angiography and the noninvasive coronary CT imaging?
  • The authors need to provide a justification for the performance of these diagnostic tests on control subjects
  • What were the criteria that were used to classify the patients as having CHD or not? This needs to be elaborated, taking into consideration the utilization of two different diagnostic methods (angiography and CT)
  • When were these studies performed? Could a subject in the control group had a coronary angiography that was negative more than ten years prior to the study, and was thus counted as a non-CHD patient? You see the point, so this should be clearly mentioned.

There are additional few things that should be pointed out and addressed as follows:

Line 37: “CHD prevalence decreases in developed countries”

This needs a supporting evidence and a reliable reference as the current sentence is not entirely accurate. CHD prevalence in developed countries has variable trends, as an example in the United States, there was a trend toward a decrease in prevalence in men and an increase in prevalence in women between 1994-98 and 1999-2004. Please be more specific and add high-quality citations.

Line 78-80: “we noted that retinal vessels are the only directly visible vessels in the body that share similar histological structure and pathological patterns with cerebral and cardiac vessels”

The cited reference discusses the overlapping features between retinal and cerebral vessels. Please add a second reference that have demonstrated the similarities between retinal and coronary vessels.

Line 94-95: “having appropriate blood pressure measures, blood glucose, blood lipids, height, and weight.”

What does the word "appropriate" in the inclusion criteria mean? This is a very nonspecific and vague term to use for parameters such as blood pressure, blood glucose, blood lipids, height, and weight. Please be more specific and indicate the exact cut-off values that were used for each parameter to include/exclude subjects.

Line 97: “or were too weak to comply with the research”

Same here, what does the phrase "too weak to comply with the research" actually mean? Need to be more specific as these inclusion and exclusion criteria are critical to understand the generalizability/applicability of the study

The discussion section is very short, and does not contribute significantly to the overall presentation of the study. Would recommend providing more detailed explanation on what the data mean. Additionally, consider providing possible mechanisms or pathways, compare your results with other studies, and discuss how your findings support or challenge the paradigm. Additionally, would point out any possible unanswered questions (what needs to be confirmed by future studies), and future directions.

Reviewer 5 Report

The manuscript is well written. The topic is interesting, although not new.  It is clear that the retina shows the small vessels and therefore, can show early changes related to atherosclerosis. The big question is how adding retinal examination add to risk estimation or diagnosis. How many additional patients changed their risk estimation when retinal examination was added to the traditional risk factors and clinical assessment.

  1. The authors are missing a recent publication (Deep-learning-based cardiovascular risk stratification using coronary artery calcium scores predicted from retinal photographs. Lancet Digit Health. 2021 May;3(5):e306-e316. doi: 10.1016/S2589-7500(21)00043-1.). in this manuscript 216,152 patients had retinal photographs and coronary calcium scores.
  2. An unclear point is about the selection of the group with coronary artery disease. “Coronary angiography or Coronary Computed Tomography were performed for coronary heart disease diagnosis”. Can it be that some patients in the group without CAD, actually had asymptomatic coronary atherosclerosis? Not every patient should undergo cath or CTA.